# Eleven key measures for monitoring general practice clinical activity during COVID-19: A retrospective cohort study using 48 million adults' primary care records in England through OpenSAFELY

**Louis Fisher[1], Helen J Curtis[1], Richard Croker[1], Milan Wiedemann[1], Victoria Speed[1], Christopher Wood[1], Andrew Brown[1], Lisa EM Hopcroft[1], Rose Higgins[1], Jon Massey[1], Peter Inglesby[1], Caroline E Morton[1], Alex J Walker[1], Jessica Morley[1], Amir Mehrkar[1], Seb Bacon[1], George Hickman[1], Orla Macdonald[2], Tom Lewis[3], Marion Wood[4], Martin Myers[5], Miriam Samuel[6], Robin Conibere[7], Wasim Baqir[4], Harpreet Sood[8], Charles Drury[9], Kiren Collison[4], Chris Bates[10], David Evans[1], Iain Dillingham[1], Tom Ward[1], Simon Davy[1], Rebecca M Smith[1], William Hulme[1], Amelia Green[1], John Parry[10], Frank Hester[10], Sam Harper[10], Jonathan Cockburn[10], Shaun O'Hanlon[11], Alex Eavis[11], Richard Jarvis[11], Dima Avramov[11], Paul Griffiths[11], Aaron Fowles[11], Nasreen Parkes[11], Brian MacKenna[1,4]\*, Ben Goldacre[1]**

[1]The Bennett Institute for Applied Data Science, Nuffield Department of Primary Care Health Sciences, University of Oxford, Oxford, United Kingdom; [2]Oxford Health Foundation Trust, Warneford Hospital, Oxford, United Kingdom; [3]Royal Devon University Healthcare NHS Foundation Trust, Barnstaple, United Kingdom; [4]NHS England, London, United Kingdom; [5]Lancashire Teaching Hospitals NHS Foundation Trust, Chorley, United Kingdom; [6]Queen Mary University of London, London, United Kingdom; [7]Beacon Medical Group, Plymouth, United Kingdom; [8]Sternhall Lane Surgery, London, United Kingdom; [9]Herefordshire and Worcestershire Health and Care NHS Trust, Worcester, United Kingdom; [10]TPP, Leeds, United Kingdom; [11]EMIS, Leeds, United Kingdom

\*For correspondence:
brian.mackenna@phc.ox.ac.uk

## Abstract

**Background:** The COVID-19 pandemic has had a significant impact on delivery of NHS care. We have developed the OpenSAFELY Service Restoration Observatory (SRO) to develop key measures of primary care activity and describe the trends in these measures throughout the COVID-19 pandemic.

**Methods:** With the approval of NHS England, we developed an open source software framework for data management and analysis to describe trends and variation in clinical activity across primary care electronic health record (EHR) data on 48 million adults.

We developed SNOMED-CT codelists for key measures of primary care clinical activity such as blood pressure monitoring and asthma reviews, selected by an expert clinical advisory group and conducted a population cohort-based study to describe trends and variation in these measures January 2019-December 2021, and pragmatically classified their level of recovery one year into the pandemic using the percentage change in the median practice level rate.

**Results:** We produced 11 measures reflective of clinical activity in general practice. A substantial drop in activity was observed in all measures at the outset of the COVID-19 pandemic. By April 2021, the median rate had recovered to within 15% of the median rate in April 2019 in six measures. The remaining measures showed a sustained drop, ranging from a 18.5% reduction in medication reviews to a 42.0% reduction in blood pressure monitoring. Three measures continued to show a sustained drop by December 2021.

**Conclusions:** The COVID-19 pandemic was associated with a substantial change in primary care activity across the measures we developed, with recovery in most measures. We delivered an open source software framework to describe trends and variation in clinical activity across an unprecedented scale of primary care data. We will continue to expand the set of key measures to be routinely monitored using our publicly available NHS OpenSAFELY SRO dashboards with near real-time data.

**Funding:** This research used data assets made available as part of the Data and Connectivity National Core Study, led by Health Data Research UK in partnership with the Office for National Statistics and funded by UK Research and Innovation (grant ref MC_PC_20058).The OpenSAFELY Platform is supported by grants from the Wellcome Trust (222097/Z/20/Z); MRC (MR/V015757/1, MC_PC-20059, MR/W016729/1); NIHR (NIHR135559, COV-LT2-0073), and Health Data Research UK (HDRUK2021.000, 2021.0157).

## Editor's evaluation

This paper presents an important effort to develop an open-source software framework for monitoring trends and variations in healthcare over time in England. They demonstrate a compelling example of how this system can track key healthcare indicators over the course of the COVID-19 pandemic. The paper will likely be mainly of interest to stakeholders in England, but could inspire the creation of similar systems in other countries.

## Introduction

The COVID-19 pandemic has significantly affected the capacity and delivery of both primary and secondary care within the NHS (*Moynihan et al., 2021*; *Shah et al., 2022*; *Mansfield et al., 2021*; *Fisher, 2023a*). We have previously described a data-driven approach to analyse, review and prioritise activity in NHS primary care in collaboration with a clinical advisory group through the establishment of the OpenSAFELY *NHS Service Restoration Observatory* (SRO) (*Curtis et al., 2022b*; *Curtis et al., 2023*). Following the first wave in March 2020, we found that some clinical activities such as cardiovascular disease assessment were not restored to near normal levels by December 2020 as was anticipated in guidance issued by NHS England in July 2020 (*NHS England, 2020*). This entailed a vast volume of data analysis, likely in excess of what could be realistically monitored by clinical and commissioning teams. Informed by this work and in collaboration with our clinical advisory group we suggested key measures of primary care clinical activity to support routine monitoring, targeted action and inform response to the COVID-19 pandemic (*Curtis et al., 2022b*).

OpenSAFELY is a secure analytics platform for electronic patient records built by our group on behalf of NHS England to deliver urgent academic and operational research during the pandemic. Using regularly updated data, it allows analysis of medical diagnoses, clinical tests, prescriptions, as well as demographic details such as age, sex, ethnicity, making detailed subgroup analysis possible. Through linkage of other data sources, it also provides information such as hospital admissions, registered deaths or COVID-19 testing data. We have previously undertaken an initial data driven approach using OpenSAFELY to uncover trends in high volume areas of primary care activity followed by expansion of this work to cover a wider range of clinical areas. This initial work was performed using data available in OpenSAFELY-TPP, covering 40% of all general practices in England. We have since extended the OpenSAFELY platform to both major EHR vendors in England, TPP and EMIS, allowing federated analyses and dashboards to be executed across the full primary care records for all patients registered at 99% of England's practices.

In this analysis we therefore set out to consolidate previous work to develop robust measures of primary care activity to describe trends and variation in these measures across 48 million adults' records available using a federated analysis in OpenSAFELY.

## Methods

### Study design

We conducted a retrospective cohort study using GP primary care EHR data from all England GP practices supplied by the EHR vendors TPP and EMIS.

### Data sharing

All data were linked, stored, and analysed securely within the OpenSAFELY platform: https://opensafely.org/. Data include pseudonymised data such as coded diagnoses, medications and physiological parameters. No free text data are included. All code is shared openly for review and re-use under MIT open license (https://github.com/opensafely/SRO-Measures, copy archived at swh:1:rev:b372319a70d7a9faf1235d4c461304c3e03817a5; *Fisher, 2023b*). Detailed pseudonymised patient data is potentially re-identifiable and therefore not shared. Aggregated data used to produce the table and figures in this manuscript are available here (TPP) and here (EMIS).

### Study population

We included all adult patients (n=48,352,770) who were alive and registered with a TPP or EMIS general practice (n=6389 practices) in England at the beginning of each month between January 2019 and December 2021. All coded events in each month for each monthly cohort were included. We also identified demographic variables for these patients including age, sex, region of their practice address, index of multiple deprivation (IMD) and ethnicity.

### Key measures of clinical activity

#### Development of key measures

In order to develop key measures of NHS clinical activity we convened a clinical advisory group made up of: front-line general practitioner and pharmacists; national clinical advisors and pathology leads; and clinical and research staff from the Bennett Institute. This group manually reviewed charts representing coding of clinical activity the development of which we have described in detail elsewhere (*Curtis et al., 2022b*; *Curtis et al., 2023*). Briefly, we used the CTV3 terminology coding hierarchy (the coding system available in OpenSAFELY-TPP at the time) to produce a large number of charts indicating variation in clinical coding activity between practices across a range of clinical areas. For each clinical area, these charts were manually reviewed by the clinical advisory group in a series of online meetings to prioritise clinical topics that would benefit from routine monitoring and targeted action. The clinical advisory group was asked to suggest key measures for each clinical area considering the following criteria: high volume usage, clinically relevant to front-line practice and whether they are more widely indicative of other problems in service delivery across the NHS (for example a decrease in records for blood tests for kidney function may be a true drop in GPs requesting these tests or it may be related to delays in laboratories processing the results).

The Bennett Institute team took these suggested measures and manually curated bespoke lists of codes (see below). Charts of the newly developed measures were then presented back to the clinical advisory group for a final review alongside a 'why it matters' text (*Table 1*), indicating why each measure is important to monitor.

This measure development process was a pragmatic one based on our experience developing measures for the OpenPrescribing platform (*Open Prescribing, 2023*), an online viewer of GP prescribing patterns with 20,000 unique users and 100 measures of clinical effectiveness, cost effectiveness and safety. These measures have been iterated over time according to feedback around clinical utility and changes in service delivery. Similarly, we anticipate that we will continue to develop and expand the measures developed here.

#### Codelists

For each key measure of activity, working with the clinical advisory group, we used the SNOMED-CT coding system, as the mandated NHS standard, to develop a single codelist (*Table 1*), which can be

**Table 1.** Development of key measures and their associated codelists.

A link to each codelist used to define the final key measure is given; all codelists are openly available for inspection and re-use at opencodelists.org.

| Suggested measure | What is it and why does it matter? | What does the measure capture? | Prior observed CTV3 code(s) | SNOMED codelist development |
|---|---|---|---|---|
| Blood pressure monitoring | A commonly-used assessment used to identify patients with hypertension or to ensure optimal treatment for those with known hypertension. This helps ensure appropriate treatment, with the aim of reducing long-term risks of complications from hypertension such as stroke, myocardial infarction and kidney disease. | Rate of blood pressure monitoring as indicated by recording of systolic blood pressure observable entities resulting from monitoring. | Codes beginning with 24: 'Examination of cardiovascular system (& [vascular system])' (*Curtis et al., 2023*) | QOF codelist for blood pressure monitoring (*NHS Digital, 2023b*), filtered to systolic codes only[†] (Codelist) |
| Cardiovascular Disease 10 Year Risk Assessment | A commonly-used risk assessment used to identify patients with an increased risk of cardiovascular events in the next 10 years. *QRISK3, 2018* This helps ensure appropriate treatment, with the aim of reducing long term risks of complications such as stroke or myocardial infarction. | Rate of cardiovascular risk assessment as indicated by a recorded code for a 10-year risk score observable entity. | XaQVY: 'QRISK2 cardiovascular disease 10-year risk score' (*Curtis et al., 2023*) | QOF codelist for all cardiovascular risk scoring tools (Codelist) |
| Cholesterol Testing | A commonly-used blood test used as part of a routine cardiovascular disease 10-year risk assessment *QRISK3, 2018* and also to identify patients with lipid disorders (e.g. familial hypercholesterolaemia). This helps ensure appropriate treatment, with the aim of reducing long term risks of complications such as stroke or myocardial infarction. | Rate of testing as indicated by a recorded code for a procedure to assess cholesterol level, observable entity returned in response to the assessment or a clinical finding associated with the result. | XE2eD: 'Serum cholesterol (& level)' (*Curtis et al., 2022b*) | 1: Converted existing CTV3 codes previously identified (using Kahootz CTV3 browser) 2: Searched SNOMED-CT for 'cholesterol' and selected any codes which related to total cholesterol monitoring/level. (Codelist) |
| Liver Function Testing - Alanine aminotransferase (ALT) | An ALT blood test is one of a group of liver function tests (LFTs) which are used to detect problems with the function of the liver. It is often used to monitor patients on medications which may affect the liver or which rely on the liver to break them down within the body. They are also tested for patients with known or suspected liver dysfunction. | Rate of testing as indicated by a recorded code for a procedure to assess ALT level or the observable entity returned in response to the assessment. | XaLJx: 'Serum alanine aminotransferase level' X77WP: 'Liver function tests' (*Curtis et al., 2022b*) | We searched SNOMED-CT for 'alanine aminotransferase' and selected all codes with reference to the test measurement/level. (Codelist) |
| Thyroid Testing - Thyroid Stimulating Hormone (TSH) | TSH is used for the diagnosis and monitoring of hypothyroidism and hyperthyroidism, including making changes to thyroid replacement therapy dosing. | Rate of testing as indicated by a recorded code for a procedure to assess TSH level or observable entity returned in response to the assessment. | XaELV: 'Serum TSH level' (*Curtis et al., 2022b*) | We searched SNOMED-CT for the term 'thyroid stimulating hormone' and selected all codes with reference to the test measurement/ level, excluding those referring to a specific timescale e.g '120 min'. (Codelist) |
| Full Blood Count - Red Blood Cell (RBC) Testing | RBC is completed as part of a group of tests referred to as a full blood count (FBC), used to detect a variety of disorders of the blood, such as anaemia and infection. | Rate of testing as indicated by a recorded code for a procedure to assess RBC count, observable entity returned in response to the assessment or a clinical finding associated with the result. | Codes beginning with 426: 'Red blood cell count' (*Curtis et al., 2022b*) | We searched for the team 'red blood cell', and included all codes relating to 'count' and excluding any sub-types of RBC testing. (Codelist) |

*Table 1 continued on next page*

*Table 1 continued*

| Suggested measure | What is it and why does it matter? | What does the measure capture? | Prior observed CTV3 code(s) | SNOMED codelist development |
|---|---|---|---|---|
| Glycated Haemoglobin Level (HbA1c) Testing | HbA1c is a long term indicator of diabetes control. NICE guidelines recommend that individuals with diabetes have their HbA1c measured at least twice a year. **NICE, 2015** Poor diabetic control can place individuals living with diabetes at an increased risk of the complications of diabetes. | Rate of testing as indicated by a recorded code for a procedure to assess HbA1c level, observable entity returned in response to the assessment or a clinical finding associated with the result. | XaPbt: 'Haemoglobin A1c level - IFCC standardised' X772q: 'Haemoglobin A1c level' (**Curtis et al., 2022b**) | 1: Converted existing CTV3 codes previously identified (using Kahootz CTV3 browser) 2: Searched for 'haemoglobin A1c' and selected any codes related to total HbA1c monitoring/ level, excluding any codes for other purposes, e.g. reference ranges. (Codelist) |
| Renal Function Assessment - Sodium Testing | Sodium is completed as part of a group of tests referred to as a renal profile, used to detect a variety of disorders of the kidneys. A renal profile is also often used to monitor patients on medications which may affect the kidneys or which rely on the kidneys to remove them from the body. | Rate of testing as indicated by a recorded code for the observable entity returned in response to an assessment. | XE2q0: 'Serum sodium level' (**Curtis et al., 2022b**) | 1: Converted existing CTV3 codes previously identified (using Kahootz CTV3 browser) 2: Searched for 'plasma sodium' and 'sodium level' 3: Limited to codes in current use, and with a numerical value within expected range* (Codelist) |
| Asthma Reviews | The British Thoracic Society and Scottish Intercollegiate Guidelines Network on the management of asthma recommend that people with asthma receive a review of their condition at least annually. If a patient has not been reviewed, it is possible that their asthma control may have worsened, leading to a greater chance of symptoms and admission to hospital. **British Thoracic Society, 2021** | Rate of reviews as indicated by a recorded code for an asthma review procedure, the regime used or the completion of an assessment. | Xaleq: 'Asthma annual review' (**Curtis et al., 2022b**) | QOF codelist (Codelist) |
| Chronic Obstructive Pulmonary Disease (COPD) Reviews | It is recommended by NICE that all individuals living with COPD have an annual review with the exception of individuals living with very severe (stage 4) COPD being reviewed at least twice a year. **NICE, 2018** If a patient has not been reviewed, it is possible that their COPD control may have worsened, leading to a greater chance of symptoms and admission to hospital. | Rate of reviews as indicated by a recorded code for the regime used. | Xalet: 'COPD review' (**Curtis et al., 2022b**) | QOF codelist (Codelist) |
| Medication Review | Many medicines are used long-term and they should be reviewed regularly to ensure they are still safe, effective and appropriate. Medication review is a broad term ranging from a notes-led review without a patient, to an in-depth Structured Medication Review with multiple appointments and follow-up. The codelist provided captures all types of reviews to give an overview of medication reviews in primary care. | Rate of recording of a code indicating medication review procedure or regime. | Various, including XaF8d: 'Medication review done' (**Curtis et al., 2023**) | QOF codelist (Codelist) |

QOF = Quality and Outcomes Framework.
*This was to avoid double counting where other codes are recorded for the testing activity alongside results being received.
†This was to avoid double counting where both systolic and diastolic codes are recorded together.

deployed across any system using SNOMED-CT. Where a well-defined nationally curated codelist existed, such as Quality and Outcomes Framework (QOF), we used the codelist released by NHS Digital (**NHS Digital, 2020a**). For pathology testing measures with no NHS mandated codelists, we searched for SNOMED-CT codes using OpenCodelists. For this proof of concept, we pragmatically

decided not to implement all additional complex logic and exceptions that may be associated with national schemes.

Codes include clinical findings (representing results of clinical observations/assessments) procedures (representing activities performed in the provision of healthcare), situation (represents concepts where the clinical concept is specified as part of the definitions, e.g. 'medication review due', or 'medication review done'), observable entities representing questions/assessments which can produce an answer/result. The codelists for each measure can contain more than one of these; the aim is to capture broad activity, which may be captured by several codes of different types. For example, to capture patients who have had a cardiovascular disease 10-year risk assessment, we have included both procedure code, which capture the act of taking the assessment, as well as codes that reflect the risk score resulting from this procedure.

### Data processing

For each measure, we calculated the monthly rate of coding activity per 1000 registered adults for each practice. Where multiple codes from a single codelist are recorded in the patient record in a single month only a single record will be returned. This is advantageous where a practice may use multiple codes to document a single broad activity carried out on a single day, for example recording *Asthma annual review* (SNOMED code 394700004) and a component of the review *Asthma medication review* (394720003) but will not capture two genuine activities carried out at a different time in the same month, for example blood tests measured 3 weeks apart.

We excluded practices not recording a single instance of a relevant code in each codelist across the entire study period from further analysis;as all the measures analysed here are high volume, any practices with zero recorded events for a measure are likely atypical. We counted the number of practices using each codelist, as well as the total number of unique patients with events across the entire study period, and the total number of events they each experienced (with each patient contributing a maximum of 1 event per month). We then calculated the median and deciles of coding activity rates across all practices each month. We present the data in time trend decile charts, which we make openly available at reports.opensafely.org and can update regularly.

### Classification of service restoration

For each key measure chart, we classified the change in coding activity using the median rate in April 2020 and 2021 compared to April 2019 which we defined as the 'baseline'. April was identified as the first full month of a full 'lockdown' in England in 2020 and additionally had 20 business days in 2019, 2020, and 2021, allowing fair comparison. The classification system (*Box 1*), extends previously developed methods to classify change from baseline based on percentage changes (*Curtis et al., 2022b*).

---

**Box 1. Service change classification relative to baseline (April 2019).**

1. For April 2020 and April 2021

   **no chang**e: activity remained within 15% of the baseline level;
   **increase**: an increase of >15% from baseline;
   **decrease**: a decrease of >15% from baseline;

2. Overall classification:

   **no change**: no change in both April 2020 and April 2021;
   **increase**: an increase in April 2020 and April 2021;
   **temporary increase**: an increase in April 2020 which returned to no change by April 2021.
   **delayed increase**: no change in April 2020 and an increase in April 2021.
   **delayed decrease**: no change in April 2020 and a decrease in April 2021.
   **sustained drop**: a decrease in April 2020 which did not return to no change by April 2021;
   **recovery**: a decrease in April 2020, which returned to no change by April 2021.

## Software and reproducibility

Data management and analysis was performed using the OpenSAFELY software libraries and Python, both implemented using Python 3.8 with all code shared openly for review and reuse github.com/opensafely/SRO-Measures. All codelists used are openly available for inspection and re-use at OpenSAFELY Codelists (*Open Codelists, 2020*). This analysis was delivered using federated analysis through the OpenSAFELY platform: codelists and code for data management and data analysis were specified once using the OpenSAFELY tools; then transmitted securely to the OpenSAFELY-TPP platform within TPP's secure environment, and separately to the OpenSAFELY-EMIS platform within EMIS's secure environment, where they were each executed separately against local patient data; summary results were then reviewed for disclosiveness, released, and combined for the final outputs. All code for the OpenSAFELY platform for data management, analysis and secure code execution is shared for review and re-use under open licenses at github.com/opensafely-core ().

## Patient and public involvement

This analysis relies on the use of large volumes of patient data. Ensuring patient, professional, and public trust is therefore of critical importance. Maintaining trust requires being transparent about the way OpenSAFELY works, and ensuring patient and public voices are represented in the design and use of the platform. Between February and July 2022 we ran a 6-month pilot of Patient and Public Involvement and Engagement activity designed to be aligned with the principles set out in the Consensus Statement on Public Involvement and Engagement with Data-Intensive Health Research (*Aitken et al., 2019*). Our engagement focused on the broader OpenSAFELY platform and comprised three sets of activities: explain and engage, involve and iterate and participate and promote. To engage and explain, we have developed a public website at opensafely.org that provides a detailed description of the OpenSAFELY platform in language suitable for a lay audience and are co-developing an accompanying explainer video. To involve and iterate, we have created the OpenSAFELY 'Digital Critical Friends' Group; comprised of approximately 12 members representative in terms of ethnicity, gender, and educational background, this group has met every 2 weeks to engage with and review the OpenSAFELY website, governance process, principles for researchers and FAQs. To participate and promote, we are conducting a systematic review of the key enablers of public trust in data-intensive research and have participated in the stakeholder group overseeing NHS England's 'data stewardship public dialogue'.

## Results

Our study included 48,352,770 registered adult patients across 6389 practices, >98% of total practices in England. A description of patient characteristics of the study population is described in *Table 2*. We developed a suite of 11 key measures indicative of clinical activity to inform restoration of NHS care in general practice, in collaboration with a clinical advisory group. These key measures include routine blood tests (cholesterol, liver function, thyroid, full blood count, glycated haemoglobin, renal function), reviews for long-term conditions (asthma, chronic obstructive pulmonary disorder [COPD], medication review), cardiovascular disease (CVD) risk assessment, and blood pressure monitoring (which may be recorded for routine monitoring or diagnosis of acute conditions). From January 2019 to December 2021 we identified 447 million recorded events across the 11 key measures.

### Study measures

For each measure, the top five most commonly used individual codes from each codelist and commonly used codes by EHR provider are presented in *Supplementary file 1*.

### Trends and variation in measures

Rates of activity for each key measure, the number of events recorded and the number of unique patients in which these events occurred across the entire study period are shown in *Table 3*. The number of patients included from January 2019 to December 2021 ranged from 1.16 million for the COPD review measure to 27.77 million patients for the blood pressure monitoring measure, representing 2.60 million and 79.30 million coded events, respectively. The median practice level rate per

**Table 2.** Cohort description using the latest recorded value for all adult patients who were registered at a general practice at any point between January 2019 and December 2021.

| Characteristic | Category | Number of adult patients (% of total population) |
|---|---|---|
| Total | | 48,352,770 (100.0) |
| | 18–19 | 1,398,430 (2.9) |
| | 20–29 | 7,685,615 (15.9) |
| | 30–39 | 8,753,520 (18.1) |
| | 40–49 | 7,754,940 (16.0) |
| | 50–59 | 8,025,250 (16.6) |
| | 60–69 | 6,316,120 (13.1) |
| | 70–79 | 5,068,760 (10.5) |
| | 80+ | 3,350,125 (6.9) |
| Age | Missing | 20 (<0.1) |
| | M | 24,002,030 (49.6) |
| Sex | F | 24,350,740 (50.4) |
| | South Asian | 3,148,455 (6.5) |
| | Black | 1,333,335 (2.8) |
| | Mixed | 604,600 (1.3) |
| | Other | 1,039,730 (2.2) |
| | White | 27,900,210 (57.7) |
| Ethnicity | Missing | 14,326,440 (29.6) |
| | Most deprived | 9,352,000 (19.3) |
| | 2 | 10,061,470 (20.8) |
| | 3 | 9,788,670 (20.2) |
| | 4 | 9,379,320 (19.4) |
| | Least deprived | 9,241,205 (19.1) |
| IMD quintile | Missing | 530,100 (1.1) |
| | East | 5,222,485 (10.8) |
| | Midlands | 8,931,820 (18.5) |
| | London | 8,499,335 (17.6) |
| | North East | 5,897,280 (12.2) |
| | North West | 7,421,095 (15.3) |
| | South East | 7,671,845 (15.9) |
| | South West | 4,706,435 (9.7) |
| Region | Missing | 2,455 (<0.1) |
| | TPP | 28,765,400 (59.5) |
| EHR provider | EMIS | 19,587,370 (40.5) |

IMD: index of multiple deprivation, EHR: electronic health record.

**Table 3.** OpenSAFELY NHS SRO Key Measures and their recorded counts and median rate of activity across practices, January 2019-December 2021.

| Key measure | Number of patients experiencing an event at least once (millions) | Number of events (millions) | Median number of coded events per 1000 registered patients in April 2019 | Median number of coded events per 1000 registered patients in April 2020 (% change vs April 2019) | Median number of coded events per 1000 registered patients in April 2021 (% change vs April 2019) | Classification (See Box 1) |
|---|---|---|---|---|---|---|
| Blood pressure monitoring | 27.77 | 79.30 | 65.03 | 9.22 (-85.82) | 37.7 (-42.03) | Sustained drop |
| Cardiovascular Disease 10-Year Risk Assessment | 7.38 | 10.49 | 6.65 | 0.61 (-90.83) | 4.14 (-37.74) | Sustained drop |
| Cholesterol Testing | 16.82 | 32.71 | 23.99 | 1.98 (-91.75) | 20.94 (-12.71) | Recovery |
| Liver Function Testing - Alanine aminotransferase (ALT) | 23.36 | 54.14 | 36.0 | 7.47 (-79.25) | 34.91 (-3.03) | Recovery |
| Thyroid Testing - Thyroid Stimulating Hormone (TSH) | 19.36 | 36.16 | 23.65 | 3.62 (-84.69) | 23.26 (-1.65) | Recovery |
| Full Blood Count - Red Blood Cell (RBC) Testing | 23.82 | 56.95 | 37.88 | 8.85 (-76.66) | 37.13 (-1.98) | Recovery |
| Glycated Haemoglobin A1c Level (HbA1c) Testing | 20.57 | 42.80 | 28.86 | 3.33 (-88.46) | 28.2 (-2.29) | Recovery |
| Renal Function Assessment - Sodium Testing | 25.07 | 65.99 | 43.88 | 9.45 (-78.46) | 41.74 (-4.88) | Recovery |
| Asthma Reviews | 3.41 | 7.15 | 3.61 | 2.17 (-39.89) | 2.76 (-23.55) | Sustained drop |
| Chronic Obstructive Pulmonary Disease (COPD) Reviews | 1.16 | 2.60 | 1.10 | 0.30 (-72.73) | 0.77 (-30.00) | Sustained drop |
| Medication Review | 22.47 | 58.27 | 34.10 | 21.68 (-36.42) | 27.80 (-18.48) | Sustained drop |

1000 registered patients at baseline ranged from 1.10 in COPD reviews to 65.03 in blood pressure monitoring.

In April 2020, for all measures the median dropped substantially compared to April 2019, ranging from a 91.75% reduction in cholesterol tests (23.99–1.98 recorded codes per 1000 registered patients) to a 36.42% reduction in medication reviews (34.10–21.68 recorded codes per 1000 registered patients; *Table 3*). By April 2021, the change in the median compared with April 2019 ranged from a decrease of 42.03% in blood pressure monitoring (April 2019: 65.03, April 2021: 37.70) to a decrease of 1.65% in thyroid testing (April 2019: 23.65, April 2021: 23.26). By April 2021 activity in all six blood monitoring measures had 'recovered' to within 15% of baseline, based on the simple SRO classification system. The remaining measures were all classified as having a 'sustained drop'. Reviews for asthma and COPD experienced reductions of 39.89% and 72.73% in 2020, respectively. These reductions were sustained in 2021 with rates of 2.76 (-23.55% from baseline) and 0.77 (-30.00% from baseline) in asthma and COPD, respectively. Blood pressure monitoring and assessment of cardiovascular 10-year risk were also classified with rates dropping by 42.03% and 37.74% between April 2019 and April 2021.

*Figure 1* shows practice level decile charts of the monthly rate per 1000 registered patients for each measure of activity; routinely updating charts are available on the OpenSAFELY Reports website (*OpenSAFELY Reports, 2022*). Most measures show a similar pattern, a steady rate with wide variation prior to the pandemic with a steep decline in April 2020 during the national lockdown, followed by partial or full recovery over the summer of 2020 and into 2021. Blood pressure monitoring, cardiovascular disease risk assessment and medication reviews continued to show a sustained drop as of December 2021. Blood tests (renal function assessment, cholesterol testing, liver function testing, thyroid function testing, full blood count and glycated haemoglobin) all show a temporary decrease in rates in September 2021. This is likely to be a consequence of a shortage of blood specimen collecting

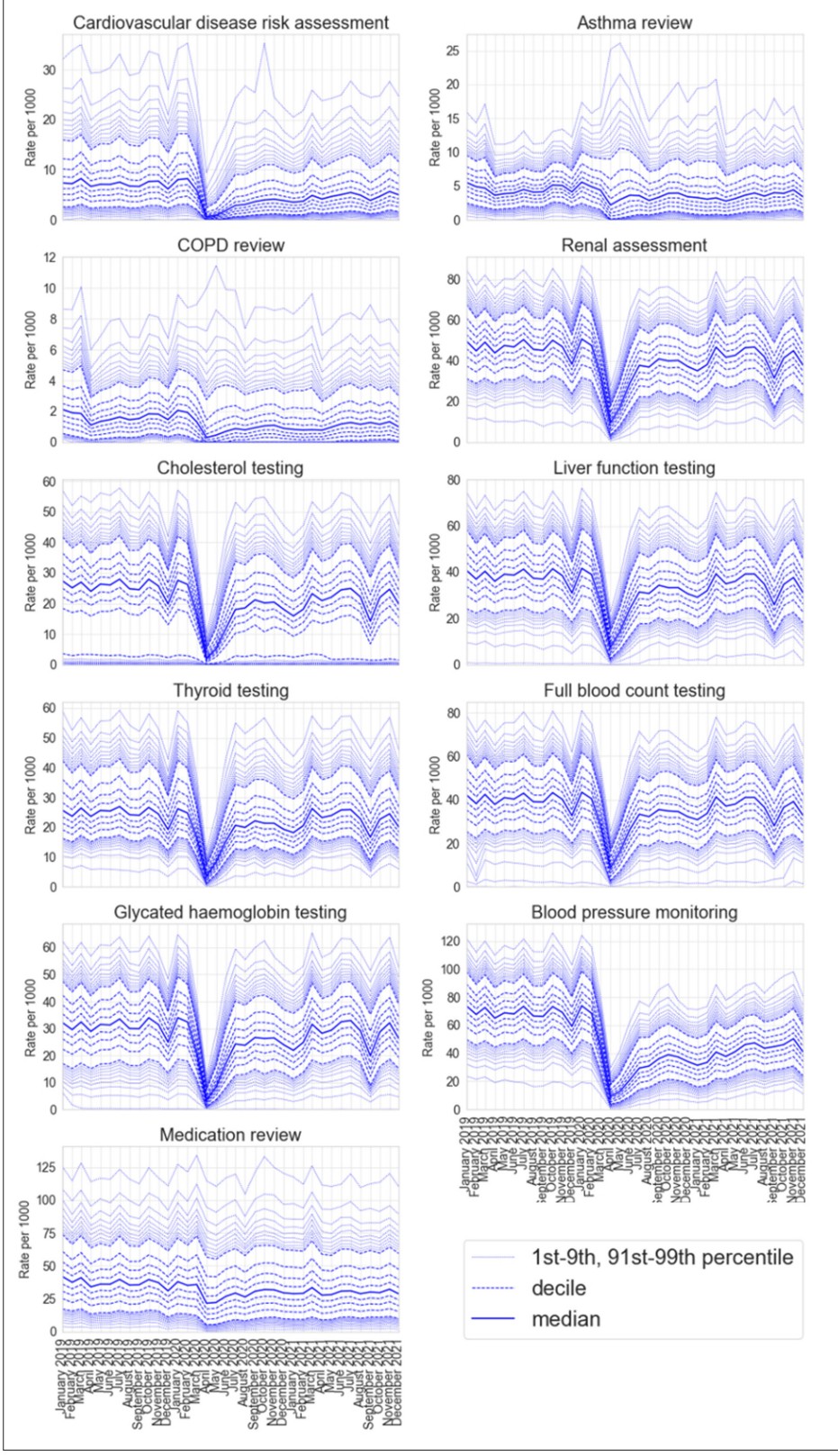

**Figure 1.** Decile charts of the practice level rate of recorded coding activity per 1000 registered patients in each identified key measure of GP activity between January 2019 and December 2021.

tubes rather than a result of the pandemic, with national guidance to temporarily halt non-clinically urgent blood tests (*DHSC & NHS England and Improvement, 2021*).

## Discussion

We have developed 11 key measures of clinical activity and using the OpenSAFELY platform, we executed a federated analysis of changes in these measures throughout the COVID-19 pandemic, across 48 million adults registered at 6389 general practices in England. These key measures demonstrated substantial changes in clinical activity. Six of the measures recovered to their pre-pandemic baseline within a year of the pandemic, showing a rapid, adaptive response by primary care in the midst of a global health pandemic.

The remaining five measures showed a more sustained drop in activity; asthma and COPD reviews did not recover to their pre-pandemic baseline until around August 2021 and blood pressure monitoring, cardiovascular disease risk assessment and medication reviews had a sustained drop in activity that persisted up to December 2021.

### Strengths and weaknesses

The key strengths of this study are the scale and completeness of the underlying raw EHR data. The OpenSAFELY platform allows federated analysis to be run across the full dataset of all raw, single-event-level clinical events for 57.9 million patients; all patients registered at 99% of all general practices in England. OpenSAFELY can provide data in near-real time, providing unprecedented opportunities for audit and feedback to rapidly identify and resolve concerns around health service activity. We choose when to update the data, and currently update on a weekly basis, meaning the the delay from occurrence of a clinical event to it appearing in the OpenSAFELY platform varies from 2 to 9 days. This is substantially faster than any other source of GP data, including those giving much less complete records. OpenSAFELY has already been used to understand disease risk (*Williamson et al., 2020*), monitor vaccination coverage (*Curtis et al., 2022a*; *Parker et al., 2023*) and novel treatments (*Green et al., 2023*), assess patient safety (*Fisher et al., 2022*), inform public health guidance and policy and much more (*NCBI, 2020*). These approaches used in these analyses are widely applicable beyond the COVID-19 pandemic.

We also recognise some limitations. With the exception of a small amount of legally restricted data, all occurrences of clinical codes are included, however coded activity may not reflect the true scale and breadth of activity. Codes recorded in general practice do not necessarily indicate unique or new events; for example one patient encounter could generate several similar codes, one patient might have similar diagnoses recorded multiple times over time, or practices might bulk-import information. For each measure, we count no more than one coded event per patient per calendar month, which avoids overcounting where practices use multiple codes to describe a single encounter, but will not account for genuine multiple encounters in a single calendar month. For some of the key measures reflecting routine testing, only test results returned to GPs are included, which will usually exclude tests requested while a person is in hospital and other settings like a private clinic. The key measures developed are not exhaustive and some important clinical areas are not covered, including mental health and female and reproductive health. We have discussed the reasons for these omissions previously (*Curtis et al., 2023*), but briefly, it can be difficult to capture clinical activity in some areas due to incomplete coding or care spanning different services such as community trusts.

Our classification system for service change is deliberately simple. A 15% window around the pre-pandemic baseline was chosen as a pragmatic cutoff to highlight these changes. We accept that recovery to the pre-pandemic baseline may not always be expected or appropriate. Finally we are only capturing key measures of clinical activity which do not reflect all clinical care carried out by practices, administrative activity, referral, liaison with other services and other services delivered by general practice.

### Findings in context

The disruption to health services as a result of the COVID-19 pandemic has been felt globally, with the WHO finding 94% of 135 countries reported some kind of disruption and 48% reported >5% disruption to primary care (*World Health Organisation, 2021*). Similarly, a systematic review of utilisation

of healthcare services during the pandemic reported a 37% reduction in services overall across p20 countries (*Moynihan et al., 2021*). A study in the UK Clinical Practice Research Datalink (CPRD) of primary care contacts for physical and mental health in the UK showed a considerable drop in activity as a result of national restrictions which only partly recovered by July 2020 (*Mansfield et al., 2021*). Despite changes in evaluated clinical activity, in the winter of 2021 NHS Digital reported that general practice delivered 34.6 million appointments representing a 26% increase (7.1 million appointments) in November 2021 compared to pre-pandemic (*NHS Digital, 2023a*).

Discussion of the specific causes and reasons for the changes in narrow measures of clinical activity we have described is outside the scope of this paper and is best addressed through quantitative analyses that identify practices in high and low deciles to approach for targeted qualitative interviews with patients and front line staff. However we believe the following broad points may help aid interpretation. Our measures reflect only a few areas of high volume clinical activity; decreases may reflect appropriate prioritisation of other clinical activity as we have found with INR tests (*Curtis et al., 2022b*) or the delivery of COVID-19 vaccinations (*NHS England, 2021b*). We have previously described how reduced clinical activity can be explained by changes in guidance and financial incentives (*Curtis et al., 2023*). For example NHS Health Checks, which are used to detect early signs of high blood pressure, heart disease or type 2 diabetes, were paused during the pandemic; this is likely to explain the sustained drop in activity in cardiovascular disease risk assessment and blood pressure monitoring (*Office for Health Improvement & Disparities, 2022*). However, in specific cases, this may reflect changes in the style of delivery of a clinical activity, rather than the volume: for example, where patients record their own blood pressure at home since, as we have previously highlighted, home monitoring of blood pressure may not be recorded completely or consistently in GP records. In addition, not all reductions should be interpreted as problematic: as part of the COVID-19 recovery, health systems are aiming to be more resilient, responsive and sustainable (*Durski et al., 2020*); complete recovery may not always be appropriate and reductions in clinical activity across some domains may reflect rational reprioritisation of activity. Where these changes in priority have not been nationally planned, data analyses such as ours may help to rapidly identify the pragmatic changes in prioritisation being made by individual dispersed organisations or people across the healthcare ecosystem before those changes are explicitly surfaced or discussed through other mechanisms.

## Policy implications and interpretation

This set of analyses has substantial implications for COVID-19 recovery specifically; the federated analytics platform and framework delivered for these analyses has substantial implications for use of GP data in service improvement and recovery. The COVID-19 pandemic has brought a new challenge for general practice to deliver safe and effective care. Our study, like previous work, has shown substantial changes in clinical activity particularly during the first English lockdown in April 2020 with a quick recovery in certain activities. The measures we have developed with our clinical advisory group are presented here as good measures of clinical activity in general practice and can be easily updated and monitored using our routinely updated dashboards on reports.opensafely.org, although we recommend that they should not be used in isolation as a sole measure of general practice activity. We can expand on these measures to include any measures needed to support NHS England's ambition to 'build back better' as we recover from the COVID-19 pandemic (*DHSC/ONS/GAD/HO, 2020*; *NHS England, 2021a*). We can update this analysis regularly with extended follow-up time and further measures of activity such as measures defined by the Quality & Outcomes Framework (QOF) and the PINCER medication safety indicators (*Fisher, 2023a*; *Avery et al., 2012*) using near-real time data to inform continued progress with NHS service restoration.

More broadly, we have developed an extendable framework for assessing primary care activity and enabling monitoring of service recovery. This framework allows fine-grained analysis over 58 million patient records; analysis that is only possible as we have developed a modular system that allows for federated analytics, where all code written for data curation and analysis is written once, and executed in different locations containing different patients' data held by different providers. Federated analytics across this scale of NHS EHR data is unprecedented. This approach is efficient: analyses can be easily updated, and expanded, because they are executed in a single framework from re-executable code. It also preserves patient trust: OpenSAFELY was the single most highly trusted COVID-19 data project in a rigorous Citizens Jury sponsored by the NHS and the National Data Guardian (*Malcolm Oswald,*

*2021*). We have also developed interactive 'point and click' infrastructure, OpenSAFELY-Interactive (https://interactive.opensafely.org/) to support delivery of dashboards and we are working with NHS England to make this tool available to approved users in order to perform their own similar analyses. OpenSAFELY access is now available to users beyond our own group and we encourage others to use the OpenSAFELY platform and the framework presented here, to develop their own measures of clinical activity. We also plan to develop the functionality for individual practices to receive near real-time feedback on the measures presented here, informing their recovery of service.

## Future work

We have previously highlighted important areas for future research following our work developing the OpenSAFELY SRO (*Curtis et al., 2023*). This includes extending the monitoring of activity to more granular groups, such as those with long-term conditions as well as assessing changes in activity in different demographic subgroups to assess health inequalities. OpenSAFELY can support these follow up analyses.

## Summary/conclusion

The COVID-19 pandemic was associated with a substantial change in healthcare activity across the measures we developed. We successfully delivered a secure open source software framework to describe trends and variation in clinical activity across an unprecedented scale of primary care data using federated analytics. We will continue to monitor these changes using our publicly available NHS OpenSAFELY SRO dashboards.

## Information governance and ethical approval

NHS England is the data controller of the NHS England OpenSAFELY COVID-19 Service; EMIS and TPP are the data processors; all study authors using OpenSAFELY have the approval of NHS England (*NHS Digital, 2020b*). This implementation of OpenSAFELY is hosted within the EMIS and TPP environments which are accredited to the ISO 27001 information security standard and are NHS IG Toolkit compliant (*NHS Digital, 2018*); Patient data has been pseudonymised for analysis and linkage using industry standard cryptographic hashing techniques; all pseudonymised datasets transmitted for linkage onto OpenSAFELY are encrypted; access to the NHS England OpenSAFELY COVID-19 service is via a virtual private network (VPN) connection; the researchers hold contracts with NHS England and only access the platform to initiate database queries and statistical models; all database activity is logged; only aggregate statistical outputs leave the platform environment following best practice for anonymisation of results such as statistical disclosure control for low cell counts (*NHS Digital, 2013*). The service adheres to the obligations of the UK General Data Protection Regulation (UK GDPR) and the Data Protection Act 2018. The service previously operated under notices initially issued in February 2020 by the the Secretary of State under Regulation 3(4) of the Health Service (Control of Patient Information) Regulations 2002 (COPI Regulations), which required organisations to process confidential patient information for COVID-19 purposes; this set aside the requirement for patient consent (*Coronavirus, 2022*). As of 1 July 2023, the Secretary of State has requested that NHS England continue to operate the Service under the COVID-19 Directions 2020 *Secretary of State for Health and Social Care - UK Government, 2020*. In some cases of data sharing, the common law duty of confidence is met using, for example, patient consent or support from the Health Research Authority Confidentiality Advisory Group (*Health Research Authority, 2023*). Taken together, these provide the legal bases to link patient datasets using the service. GP practices, which provide access to the primary care data, are required to share relevant health information to support the public health response to the pandemic, and have been informed of how the service operates.

## Acknowledgements

We are very grateful for all the support received from TPP and EMIS throughout this work, and for generous assistance from the information governance and database teams at NHS England and the NHS England Transformation Directorate. Funding This research used data assets made available as part of the Data and Connectivity National Core Study, led by Health Data Research UK in partnership with the Office for National Statistics and funded by UK Research and Innovation (grant ref MC_PC_20058). In addition, the OpenSAFELY Platform is supported by grants from the Wellcome

Trust (222097/Z/20/Z); MRC (MR/V015757/1, MC_PC-20059, MR/W016729/1); NIHR (NIHR135559, COV-LT2-0073), and Health Data Research UK (HDRUK2021.000, 2021.0157). BG has also received funding from: the Bennett Foundation, the Wellcome Trust, NIHR Oxford Biomedical Research Centre, NIHR Applied Research Collaboration Oxford and Thames Valley, the Mohn-Westlake Foundation; all Bennett Institute staff are supported by BG's grants on this work. BMK is employed by NHS England and seconded to the Bennett Institute. The views expressed are those of the authors and not necessarily those of the NIHR, NHS England, UK Health Security Agency (UKHSA) or the Department of Health and Social Care. Funders had no role in the study design, collection, analysis, and interpretation of data; in the writing of the report; and in the decision to submit the article for publication.

## Additional information

### Competing interests

Amir Mehrkar: Former employee and interim CMO of NHS Digital. Member of the RCGP health informatics group and the NHS digital GP data Professional Advisory Group. Orla Macdonald: Received payment for:- Presenter at Psych 1 courses (College of Mental Health Pharmacy)- External assessor for Aston University modules on the PG Dip Psych Pharm course.Member of the College of Mental Health Pharmacy Council and co-lead the Education portfolio for this council. This is a charitable organisation and my work for the council is not paid, however, expenses for travel are reimbursed. Martin Myers: Chair of MHRA IVD Expert Advisory Group. Unpaid. Miriam Samuel: Between August 2018 and September 2022, employed as an NIHR funded Academic Clinical Fellow in Primary Care based at Queen Mary University London.Since September 2022, employed as a clinical research fellow on the NIHR funded AI MULTIPLY grant. All funding is awarded through Queen Mary University London. Robin Conibere: Honoraria for charing/presenting on webinars with Primary Care Pharmacy Association. Chris Bates: Chris Bates is affiliated with TPP. The author has no financial interests to declare. John Parry: John Parry is affiliated with TPP. The author has no financial interests to declare. Frank Hester: Frank Hester is affiliated with TPP. The author has no financial interests to declare. Sam Harper: Sam Harper is affiliated with TPP. The author has no financial interests to declare. Jonathan Cockburn: Jonathan Cockburn is affiliated with TPP. The author has no financial interests to declare. Shaun O'Hanlon: Shaun O'Hanlon is affiliated with EMIS. The author has no financial interests to declare. Alex Eavis: Alex Eavis is affiliated with EMIS. Sits on the advisory board for OpenSAFELY and QResearch; both unpaid. Richard Jarvis: Richard Jarvis is affiliated with EMIS. The author has no financial interests to declare. Dima Avramov: Dima Avramov is affiliated with EMIS. The author has no financial interests to declare. Paul Griffiths: Paul Griffiths is affiliated with EMIS. The author has no financial interests to declare. Aaron Fowles: Aaron Fowles is affiliated with EMIS. The author has no financial interests to declare. Nasreen Parkes: Nasreen Parkes is affiliated with EMIS. The author has no financial interests to declare. Brian MacKenna: Seconded to the Bennett Institute.Trustee of ICAP, a charity delivering counselling. Ben Goldacre: BG has received research funding from the Laura and John Arnold Foundation, the NHS National Institute for Health Research (NIHR), the NIHR School of Primary Care Research, NHS England, the NIHR Oxford Biomedical Research Centre, the Mohn-Westlake Foundation, NIHR Applied Research Collaboration Oxford and Thames Valley, the Wellcome Trust, the Good Thinking Foundation, Health Data Research UK, the Health Foundation, the World Health Organisation, UKRI MRC, Asthma UK, the British Lung Foundation, and the Longitudinal Health and Wellbeing strand of the National Core Studies programme; he is a Non-Executive Director at NHS Digital; he also receives personal income from speaking and writing for lay audiences on the misuse of science. The other authors declare that no competing interests exist.

## Funding

| Funder | Grant reference number | Author |
| --- | --- | --- |
| UK Research and Innovation | MC_PC_20058 | Louis Fisher<br>Helen J Curtis<br>Richard Croker<br>Milan Wiedemann<br>Victoria Speed<br>Christopher Wood<br>Andrew Brown<br>Lisa EM Hopcroft<br>Rose Higgins<br>Jon Massey<br>Peter Inglesby<br>Caroline E Morton<br>Alex J Walker<br>Jessica Morley<br>Amir Mehrkar<br>Seb Bacon<br>George Hickman<br>Orla Macdonald<br>David Evans<br>Iain Dillingham<br>Tom Ward<br>Simon Davy<br>Rebecca M Smith<br>William Hulme<br>Amelia Green<br>Brian MacKenna<br>Ben Goldacre |
| Wellcome Trust | 222097/Z/20/Z | Helen J Curtis<br>Richard Croker<br>Milan Wiedemann<br>Victoria Speed<br>Christopher Wood<br>Andrew Brown<br>Lisa EM Hopcroft<br>Rose Higgins<br>Jon Massey<br>Peter Inglesby<br>Caroline E Morton<br>Alex J Walker<br>Jessica Morley<br>Amir Mehrkar<br>Seb Bacon<br>George Hickman<br>Orla Macdonald<br>David Evans<br>Iain Dillingham<br>Tom Ward<br>Simon Davy<br>Rebecca M Smith<br>William Hulme<br>Amelia Green<br>Brian MacKenna<br>Ben Goldacre |

| Funder | Grant reference number | Author |
|---|---|---|
| Medical Research Council | MR/V015757/1 | Helen J Curtis<br>Richard Croker<br>Milan Wiedemann<br>Victoria Speed<br>Christopher Wood<br>Andrew Brown<br>Lisa EM Hopcroft<br>Rose Higgins<br>Jon Massey<br>Peter Inglesby<br>Caroline E Morton<br>Alex J Walker<br>Jessica Morley<br>Amir Mehrkar<br>Seb Bacon<br>George Hickman<br>Orla Macdonald<br>David Evans<br>Iain Dillingham<br>Tom Lewis<br>Simon Davy<br>Rebecca M Smith<br>William Hulme<br>Amelia Green<br>Ben Goldacre<br>Brian MacKenna |
| National Institute for Health and Care Research | NIHR135559 | Helen J Curtis<br>Richard Croker<br>Milan Wiedemann<br>Victoria Speed<br>Christopher Wood<br>Andrew Brown<br>Lisa EM Hopcroft<br>Rose Higgins<br>Jon Massey<br>Peter Inglesby<br>Caroline E Morton<br>Alex J Walker<br>Jessica Morley<br>Amir Mehrkar<br>Seb Bacon<br>George Hickman<br>Orla Macdonald<br>David Evans<br>Iain Dillingham<br>Tom Lewis<br>Simon Davy<br>Rebecca M Smith<br>William Hulme<br>Amelia Green<br>Brian MacKenna<br>Ben Goldacre |

| Funder | Grant reference number | Author |
|---|---|---|
| Health Data Research UK | HDRUK2021.000 | Helen J Curtis<br>Richard Croker<br>Milan Wiedemann<br>Victoria Speed<br>Christopher Wood<br>Andrew Brown<br>Lisa EM Hopcroft<br>Rose Higgins<br>Jon Massey<br>Peter Inglesby<br>Caroline E Morton<br>Alex J Walker<br>Jessica Morley<br>Amir Mehrkar<br>Seb Bacon<br>George Hickman<br>Orla Macdonald<br>David Evans<br>Iain Dillingham<br>Tom Lewis<br>Simon Davy<br>Rebecca M Smith<br>William Hulme<br>Brian MacKenna<br>Ben Goldacre |
| NIHR Oxford Biomedical Research Centre | | Ben Goldacre |
| Mohn Westlake Foundation | | Ben Goldacre |
| Bennett Foundation | | Ben Goldacre |
| NIHR Applied Research Collaboration Oxford and Thames Valley | | Ben Goldacre |
| Wellcome Trust | | Ben Goldacre |
| Medical Research Council | MR/W016729/1 | Helen J Curtis<br>Richard Croker<br>Milan Wiedemann<br>Victoria Speed<br>Christopher Wood<br>Andrew Brown<br>Lisa EM Hopcroft<br>Rose Higgins<br>Jon Massey<br>Peter Inglesby<br>Caroline E Morton<br>Alex J Walker<br>Jessica Morley<br>Amir Mehrkar<br>Seb Bacon<br>George Hickman<br>Orla Macdonald<br>David Evans<br>Iain Dillingham<br>Tom Lewis<br>Simon Davy<br>Rebecca M Smith<br>William Hulme<br>Amelia Green<br>Brian MacKenna<br>Ben Goldacre |

| Funder | Grant reference number | Author |
|---|---|---|
| National Institute for Health and Care Research | COV-LT2-0073 | Helen J Curtis<br>Richard Croker<br>Milan Wiedemann<br>Victoria Speed<br>Christopher Wood<br>Andrew Brown<br>Lisa EM Hopcroft<br>Rose Higgins<br>Jon Massey<br>Peter Inglesby<br>Caroline E Morton<br>Alex J Walker<br>Jessica Morley<br>Amir Mehrkar<br>Seb Bacon<br>George Hickman<br>Orla Macdonald<br>David Evans<br>Iain Dillingham<br>Tom Lewis<br>Simon Davy<br>Rebecca M Smith<br>William Hulme<br>Amelia Green<br>Brian MacKenna<br>Ben Goldacre |
| Medical Research Council | MC_PC-20059 | Louis Fisher<br>Helen J Curtis<br>Richard Croker<br>Milan Wiedemann<br>Victoria Speed<br>Christopher Wood<br>Andrew Brown<br>Lisa EM Hopcroft<br>Rose Higgins<br>Jon Massey<br>Peter Inglesby<br>Caroline E Morton<br>Alex J Walker<br>Jessica Morley<br>Amir Mehrkar<br>Seb Bacon<br>George Hickman<br>Orla Macdonald<br>David Evans<br>Iain Dillingham<br>Tom Lewis<br>Simon Davy<br>Rebecca M Smith<br>William Hulme<br>Amelia Green<br>Brian MacKenna<br>Ben Goldacre |

| Funder | Grant reference number | Author |
|---|---|---|
| Health Data Research UK | 2021.0157 | Louis Fisher<br>Helen J Curtis<br>Richard Croker<br>Victoria Speed<br>Milan Wiedemann<br>Christopher Wood<br>Andrew Brown<br>Lisa EM Hopcroft<br>Rose Higgins<br>Jon Massey<br>Peter Inglesby<br>Caroline E Morton<br>Alex J Walker<br>Jessica Morley<br>Amir Mehrkar<br>Seb Bacon<br>George Hickman<br>Orla Macdonald<br>David Evans<br>Iain Dillingham<br>Tom Lewis<br>Simon Davy<br>Rebecca M Smith<br>William Hulme<br>Amelia Green<br>Ben Goldacre<br>Brian MacKenna |

The funders had no role in study design, data collection and interpretation, or the decision to submit the work for publication. For the purpose of Open Access, the authors have applied a CC BY public copyright license to any Author Accepted Manuscript version arising from this submission.

## Author contributions

Louis Fisher, Helen J Curtis, Richard Croker, Brian MacKenna, Conceptualization, Formal analysis, Investigation, Visualization, Methodology, Writing – original draft, Writing – review and editing; Milan Wiedemann, Investigation, Methodology, Writing – review and editing; Victoria Speed, Christopher Wood, Andrew Brown, Writing – original draft, Writing – review and editing; Lisa EM Hopcroft, Rose Higgins, Alex J Walker, Orla Macdonald, Tom Lewis, Marion Wood, Martin Myers, Miriam Samuel, Robin Conibere, Wasim Baqir, Harpreet Sood, Charles Drury, Kiren Collison, William Hulme, Amelia Green, Methodology, Writing – review and editing; Jon Massey, Investigation, Writing – review and editing; Peter Inglesby, Seb Bacon, George Hickman, Chris Bates, David Evans, Iain Dillingham, Tom Ward, Simon Davy, Rebecca M Smith, Sam Harper, Jonathan Cockburn, Shaun O'Hanlon, Alex Eavis, Richard Jarvis, Dima Avramov, Paul Griffiths, Aaron Fowles, Nasreen Parkes, Resources, Data curation, Software, Writing – review and editing; Caroline E Morton, Conceptualization, Resources, Data curation, Software, Investigation, Methodology, Writing – review and editing; Jessica Morley, Project administration, Writing – review and editing; Amir Mehrkar, Resources, Data curation, Supervision, Project administration, Writing – review and editing; John Parry, Resources, Data curation, Writing – review and editing; Frank Hester, Resources, Data curation, Funding acquisition, Writing – review and editing; Ben Goldacre, Conceptualization, Supervision, Funding acquisition, Writing – review and editing

## Author ORCIDs

Louis Fisher ⓘ http://orcid.org/0000-0002-0295-3812
Christopher Wood ⓘ http://orcid.org/0000-0001-5184-0606
Rose Higgins ⓘ http://orcid.org/0000-0002-5295-4370
Jon Massey ⓘ http://orcid.org/0000-0002-2497-4040
George Hickman ⓘ http://orcid.org/0000-0002-6691-4046
William Hulme ⓘ http://orcid.org/0000-0002-9162-4999
Amelia Green ⓘ http://orcid.org/0000-0002-7246-2074
Brian MacKenna ⓘ http://orcid.org/0000-0002-3786-9063

## Ethics

Human subjects: The OpenSAFELY research platform adheres to the obligations of the UK General Data Protection Regulation (GDPR) and the Data Protection Act 2018. In March 2020, the Secretary of State for Health and Social Care used powers under the UK Health Service (Control of Patient Information) Regulations 2002 (COPI) to require organisations to process confidential patient information for the purposes of protecting public health, providing healthcare services to the public and monitoring and managing the COVID-19 outbreak and incidents of exposure; this sets aside the requirement for patient consent. Taken together, these provide the legal bases to link patient datasets on the OpenSAFELY platform. GP practices, from which the primary care data are obtained, are required to share relevant health information to support the public health response to the pandemic, and have been informed of the OpenSAFELY analytics platform. This study was approved by the Health Research Authority (REC reference 20/LO/0651).

## Decision letter and Author response

Decision letter https://doi.org/10.7554/eLife.84673.sa1
Author response https://doi.org/10.7554/eLife.84673.sa2

---

# Additional files

## Supplementary files

• MDAR checklist

• Supplementary file 1. Counts of the five most commonly used codes within each measure codelist between January 2019 and April 2021.

## Data availability

All data were linked, stored and analysed securely within the OpenSAFELY platform: https://opensafely.org/. Data include pseudonymised data such as coded diagnoses, medications and physiological parameters. No free text data are included. All code is shared openly for review and re-use under MIT open license (https://github.com/opensafely/sro-measures, copy archived at *Fisher, 2023b*). Access to the underlying identifiable and potentially re-identifiable pseudonymised electronic health record data is tightly governed by various legislative and regulatory frameworks, and restricted by best practice. Access is controlled via a project application process described here (https://www.opensafely.org/onboarding-new-users/). All code for the full data management pipeline-from raw data to completed results for this analysis is available at https://github.com/opensafely/sro-measures. All code for the OpenSAFELY platform as a whole is available for review at https://github.com/opensafely-core. All aggregated outputs used in the manuscript are openly available at https://jobs.opensafely.org/datalab/service-restoration-observatory/sro-measures/outputs/ and https://jobs.opensafely.org/datalab/service-restoration-observatory/sro-measures-emis/outputs.

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
