## [Editor Report]

This paper presents an important effort to develop an open-source software framework for monitoring trends and variations in healthcare over time in England. They demonstrate a compelling example of how this system can track key healthcare indicators over the course of the COVID-19 pandemic. The paper will likely be mainly of interest to stakeholders in England, but could inspire the creation of similar systems in other countries.

---

## [Decision Letter]

**Decision letter after peer review:**

Thank you for submitting your article "Eleven key measures for monitoring general practice clinical activity during COVID-19: A retrospective cohort study using 48 million adults' primary care records in England through OpenSAFELY" for consideration by *eLife*. Your article has been reviewed by 3 peer reviewers, one of whom is a member of our Board of Reviewing Editors, and the evaluation has been overseen by a Senior Editor.

As is customary in *eLife*, the reviewers have discussed their critiques with one another. What follows below is the Reviewing Editor's edited compilation of the essential and ancillary points provided by reviewers in their critiques and in their interaction post-review. Please submit a revised version that addresses these concerns directly. Although we expect that you will address these comments in your response letter, we also need to see the corresponding revision clearly marked in the text of the manuscript. Some of the reviewers' comments may seem to be simple queries or challenges that do not prompt revisions to the text. Please keep in mind, however, that readers may have the same perspective as the reviewers. Therefore, it is essential that you attempt to amend or expand the text to clarify the narrative accordingly.

Essential revisions:

1. Add to the discussion more on the types of future analyses and data needs this platform could be used for.

2. Explain how this work differs from earlier work using the OpenSAFELY platform, as the novelty was not clear to reviewers.

2. Expand on whether the platform can be used to explore trends in specific populations, including vulnerable populations.

3. Comment on the gap in indicators identified by reviewer #3 (mental health, maternal or child health).

4. More thoroughly describe the underlying clinical activity which is intended to be measured by each code.

5. Comment on whether any validation exercise was undertaken to validate whether selected codes identified the target clinical or laboratory measurements.

*Reviewer #1 (Recommendations for the authors):*

No particular concerns about methodology.

The Results section is a little light, I would have liked to see more analyses of the data, while I admire the extensive work that went into developing this system, currently, it looks like the type of information the system can generate is not extensive. It would be nice to add more to the discussion about the types of future analyses and data needs this platform could be used for.

*Reviewer #2 (Recommendations for the authors):*

The authors only reported overall trends, without consideration of different (vulnerable) patient groups. It would be meaningful to analyse whether there were (and still are) certain populations that experienced a higher decrease in clinical activity due to the COVID-19 pandemic, in order to issue targeted initiatives.

*Reviewer #3 (Recommendations for the authors):*

Abstract: the abstract talks about "11 key measures of primary care clinical activity" but there are no examples given, so it is difficult to get a sense of what is actually being measured without reading the full paper. I suggest including some examples of the measures.

Background: as in the abstract, it would be helpful to state some examples of 'clinical activity'.

Key measures of clinical activity: the manuscript talks about CTV3 and SNOMED CT codes but it would be helpful to start off by describing the underlying clinical activity which is intended to be measured. For example, measurement of frequency of diagnosis codes may be a measure of patients seen with a particular condition or diagnosis coding activity (i.e. the proportion of diagnoses that are recorded in coded data rather than free text). Prescribing and laboratory investigation measures are more likely to measure actual clinical activity as these are almost completely recorded electronically.

The authors could also consider the use of the SNOMED CT hierarchy to generate codelists using knowledge of which codes are subtypes of others. This would simplify the description of the codelist and would ensure that future codes are automatically included if they are in an appropriate place in the hierarchy.

Table 1: SNOMED CT coded events can be of a variety of semantic types including records of diagnoses, investigation results, prescriptions etc. The table should state the clinical activity that is intended to be measured, and then the way this is represented and detected in the GP systems (i.e. TPP and EMIS), rather than lumping together as 'SNOMED CT codes'. This is because other (existing or future) GP systems may record such information differently (e.g. in Vision each SNOMED CT coded entry has an 'entity type' which denotes the type of information that is contained within that entry, so there is a distinction between e.g. BP measurements and BP diagnoses). I would expect that validation has occurred to ensure that the SNOMED CT codes used to record common clinical or laboratory measurements are those that would be expected.

---

## [Author Response]

Essential revision:1. Add to the discussion more on the types of future analyses and data needs this platform could be used for.

OpenSAFELY has already been used to support a wide range of research. We have added a summary of this research to the Discussion section which highlights the types of analyses OpenSAFELY can be used for.

2. Explain how this work differs from earlier work using the OpenSAFELY platform, as the novelty was not clear to reviewers.

This work extends on our previous work by developing a set of key measures of primary care activity to assess changes in primary care activity throughout the pandemic and extending analysis of these measures to cover >99% of the population of England. We have edited the background section to make this clearer.

2. Expand on whether the platform can be used to explore trends in specific populations, including vulnerable populations.

OpenSAFELY allows access to the full pseudonymised primary care records for patients in England. This, combined with linkage to other data sources allows for detailed subgroup analysis. Such analysis was out of scope here, but we have added detail to the background outlining the data available using OpenSAFELY and the types of analysis it allows. We have highlighted this as an important area for future research in a newly added ‘Future Research’ sub-section.

3. Comment on the gap in indicators identified by reviewer #3 (mental health, maternal or child health).

We have discussed these gaps in our previous work. Careful analysis outside the scope of this manuscript is needed to properly capture activity in these areas. We have added a section in the study weaknesses to indicate the omission of these areas as well as reasons for their omission.

4. More thoroughly describe the underlying clinical activity which is intended to be measured by each code.

A column has been added to Table 1 to outline what clinical activity each measure captures. Detail has been added to supplementary table 1 to indicate the top level SNOMED-CT hierarchy each of the most commonly used code within each measure belongs to (highlighted in yellow). This gives an indication of the clinical activity captured. A paragraph has been added to the “Codelists” section of the methods to highlight the range of clinical activity intended to be captured by the measures.

5. Comment on whether any validation exercise was undertaken to validate whether selected codes identified the target clinical or laboratory measurements.

We have confirmed the codes used here are those that would be expected through our work with general practitioners and pathology leads.

Reviewer #1 (Recommendations for the authors):No particular concerns about methodology.The Results section is a little light, I would have liked to see more analyses of the data, while I admire the extensive work that went into developing this system, currently, it looks like the type of information the system can generate is not extensive. It would be nice to add more to the discussion about the types of future analyses and data needs this platform could be used for.

We have added detail of the types data available in OpenSAFELY analysis and the wide range of analyses this has already supported. See response to public review for detail.

Reviewer #2 (Recommendations for the authors):The authors only reported overall trends, without consideration of different (vulnerable) patient groups. It would be meaningful to analyse whether there were (and still are) certain populations that experienced a higher decrease in clinical activity due to the COVID-19 pandemic, in order to issue targeted initiatives.

We have previously highlighted this as an important area of future research 7. This needs careful analysis that is outside of the scope of this manuscript, but we have acknowledged this as an important follow up analysis in a “Future Work” section added to the discussion.

Reviewer #3 (Recommendations for the authors):Abstract: the abstract talks about "11 key measures of primary care clinical activity" but there are no examples given, so it is difficult to get a sense of what is actually being measured without reading the full paper. I suggest including some examples of the measures.

An indication of the type of activity captured, with examples of measures has been included in the abstract.

Background: as in the abstract, it would be helpful to state some examples of 'clinical activity'.

An example of clinical activity has been added to the background.

Key measures of clinical activity: the manuscript talks about CTV3 and SNOMED CT codes but it would be helpful to start off by describing the underlying clinical activity which is intended to be measured. For example, measurement of frequency of diagnosis codes may be a measure of patients seen with a particular condition or diagnosis coding activity (i.e. the proportion of diagnoses that are recorded in coded data rather than free text). Prescribing and laboratory investigation measures are more likely to measure actual clinical activity as these are almost completely recorded electronically.

Detail has been added to the methods to describe the clinical activity intended to be captured. We have added a column to Table 1 to indicate the clinical activity captured by each measure.

The authors could also consider the use of the SNOMED CT hierarchy to generate codelists using knowledge of which codes are subtypes of others. This would simplify the description of the codelist and would ensure that future codes are automatically included if they are in an appropriate place in the hierarchy.

Using OpenCodelists, it is possible to see the top level hierarchy each individual code belongs to. This indicates for example, whether a code captures an observable value from an assessment or the act of carrying out a procedure. The majority of measures include codes of different types. This is deliberate. Recorded codes may be incomplete, so by including related codes that capture different stages of the pathway, it increases the chances we capture the activity we want (ie if a patient only has a code related to the procedure for taking a TSH measurement, but not the returned level recorded, they will still be captured). This point has been made clearer in the methods section of the manuscript.

Utilising the SNOMED-CT hierarchy is sensible. This hierarchy is captured by the “codelist builder” on OpenCodelists, so any future updates to SNOMED-CT can easily be incorporated into the codelists developed here. It is however important to note that automated inclusion of all codes within a sub hierarchy can result in the inclusion of unwanted codes. An example is “Systolic blood pressure of neonate at birth” (708502007) within the hierarchy of “Systolic blood pressure” (271649006). In our codelist for systolic blood pressure, which intends to capture the assessment of blood pressure within primary care, inclusion of this code is not appropriate.

Table 1: SNOMED CT coded events can be of a variety of semantic types including records of diagnoses, investigation results, prescriptions etc. The table should state the clinical activity that is intended to be measured, and then the way this is represented and detected in the GP systems (i.e. TPP and EMIS), rather than lumping together as 'SNOMED CT codes'. This is because other (existing or future) GP systems may record such information differently (e.g. in Vision each SNOMED CT coded entry has an 'entity type' which denotes the type of information that is contained within that entry, so there is a distinction between e.g. BP measurements and BP diagnoses). I would expect that validation has occurred to ensure that the SNOMED CT codes used to record common clinical or laboratory measurements are those that would be expected.

See comments discussing the addition to Table 1, the supplementary results and methods sections.

We have confirmed the codes used here are those that would be expected through our work with general practitioners and pathology leads.

1. Williamson, E. J. *et al.* Factors associated with COVID-19-related death using OpenSAFELY. *Nature* 584, 430–436 (2020).

2. Trends and clinical characteristics of 57.9 million COVID-19 vaccine recipients: a federated analysis of patients’ primary care records in situ using OpenSAFELY | British Journal of General Practice. https://bjgp.org/content/early/2021/11/08/BJGP.2021.0376.

3. Parker, E. P. *et al.* Factors associated with COVID-19 vaccine uptake in people with kidney disease: an OpenSAFELY cohort study. *BMJ Open* 13, e066164 (2023).

4. Green, A. C. A. *et al.* Trends, variation, and clinical characteristics of recipients of antiviral drugs and neutralising monoclonal antibodies for covid-19 in community settings: retrospective, descriptive cohort study of 23.4 million people in OpenSAFELY. *BMJ Med.* 2, (2023).

5. Collaborative, T. O. *et al.* Potentially inappropriate prescribing of DOACs to people with mechanical heart valves: a federated analysis of 57.9 million patients’ primary care records in situ using OpenSAFELY. 2021.07.27.21261136 https://www.medrxiv.org/content/10.1101/2021.07.27.21261136v1 (2021) doi:10.1101/2021.07.27.21261136.

6. OpenSAFELY Pubmed search results. *PubMed* https://pubmed.ncbi.nlm.nih.gov/?term=OpenSAFELY.

7. Curtis, H. J. *et al.* OpenSAFELY NHS Service Restoration Observatory 2: changes in primary care clinical activity in England during the COVID-19 pandemic. *Br. J. Gen. Pract.* 73, e318–e331 (2023).

8. OpenSAFELY NHS Service Restoration Observatory 2: changes in primary care activity across six clinical areas during the COVID-19 pandemic | medRxiv. https://www.medrxiv.org/content/10.1101/2022.06.01.22275674v1.

9. Suboptimal prescribing behaviour associated with clinical software design features: a retrospective cohort study in English NHS primary care | British Journal of General Practice. https://bjgp.org/content/70/698/e636.